# Learning Multi-Agent Communication through Structured Attentive Reasoning

**Murtaza Rangwala**
Virginia Tech
Blacksburg, VA 24060
murtazar@vt.edu

**Ryan Williams**
Virginia Tech
Blacksburg, VA 24060
rywilli1@vt.edu

## Abstract

Learning communication via deep reinforcement learning has recently been shown to be an effective way to solve cooperative multi-agent tasks. However, learning *which* communicated information is beneficial for each agent's decision-making process remains a challenging task. In order to address this problem, we explore relational reinforcement learning which leverages attention-based networks to learn efficient and interpretable relations between entities. On the foundation of relations, we introduce a novel communication architecture that exploits a memory-based attention network that selectively reasons about the value of information received from other agents while considering its past experiences. Specifically, the model communicates by first computing the relevance of messages received from other agents and then extracts task-relevant information from memories given the newly received information. We empirically demonstrate the strength of our model in cooperative and competitive multi-agent tasks, where inter-agent communication and reasoning over prior information substantially improves performance compared to baselines. We further show in the accompanying videos and experimental results that the agents learn a sophisticated and diverse set of cooperative behaviors to solve challenging tasks, both for discrete and continuous action spaces using on-policy and off-policy gradient methods. By developing an explicit architecture that is targeted towards communication, our work aims to open new directions to overcome important challenges in multi-agent cooperation through learned communication.

Code available at: https://github.com/caslab-vt/SARNet

## 1 Introduction

Communication is one of the fundamental building blocks for cooperation in multi-agent systems. Indeed, the ability to effectively represent and communicate information valuable to a task is especially important in multi-agent deep reinforcement learning (MADRL). Apart from learning what to communicate, it is critical that *agents learn to effectively reason based on the information communicated to them by their teammates*. Such a capability enables agents to develop sophisticated coordination strategies that would be invaluable in application scenarios such as search-and-rescue for multi-robot systems [1], swarming and flocking with adversaries [2], multiplayer games such as StarCraft [3] and DoTA [4], and autonomous vehicle planning [5].

In this work, we explore the concept of building agents that can solve complex cooperative tasks by answering the question: *how do agents learn to effectively communicate and reason in support of intelligent cooperation?* Humans naturally inspire such a question as they exhibit complex collaboration strategies through a structured reasoning process [6–8], allowing them to recognize, communicate, and exploit important task information. In the context of multi-agent cooperation,

we draw inspiration from work in soft-attention [9] to implement a method for computing *relations* between agents, coupled with a memory-based attention network from Compositional Attention Networks (MAC) [10], yielding a framework for communication that performs *structured attentive reasoning* over communicated information and past memories.

Concretely, we develop a communication architecture for MADRL by leveraging the approach of relational reinforcement learning (RRL) [11] coupled with the capacity to learn from past experiences. Our architecture is guided by the notion that a structured and iterative reasoning between non-local entities will enable agents to capture higher-order relations that are necessary for complex problem-solving. To achieve manageable computational complexity with variable team sizes, we exploit an adaptation of soft-attention [9] as the base operation for selectively attending to an entity or information in a given task. In an effort to better equip agents to make deliberate decisions, we separate *attention* and *reasoning* into distinct stages. Specifically, an attention unit informs the agent of which entities are most important for the current time-step, while the reasoning stage uses previous memories and guidance from the attention unit to extract the shared information that is most relevant. This explicit separation in communication enables agents to not only place importance on new information from other agents, but to selectively reason over information from past memories given new information. This communication framework is learned in an end-to-end fashion, without resorting to any supervision, as a result of task-specific rewards.

The main contributions of this work can be summarized as follows:

1. We propose a new communication architecture, the **Structured Attentive Reasoning Network (SARNet)**, where agents extract the relevance of other agents' information and reason over received communications and past memories before performing an action.

2. We show that our framework learns richer and more complex behaviors for a given task by reasoning over communicated information and past memories.

3. We introduce an extension to the twin-delayed deep deterministic policy gradient (TD3) method [12], allowing for trajectory-based training in recurrent networks for cooperative and competitive scenarios.

4. We conduct benchmarks on a mixture of discrete and continuous actions spaces, with limited or zero agent vision, using both REINFORCE [13] and our improved version of TD3 [12] to compare our approaches to relevant baselines. Our empirical study demonstrates the effectiveness of our novel architecture to solve cooperative and competitive multi-agent tasks with varying team sizes and environments.

## 2   Related Work

Communication in multi-agent deep reinforcement learning (MADRL) was formalized by CommNet [14], which shares hidden state representations among agents to augment the information available for each agent to process through their respective encoders. This framework has also been adopted by several recent works in MADRL including [15] and [16]. However, there are two shortcomings of the CommNet approach to communication. The first is that messages communicated by the agents are not concurrently used for action prediction at the current time step, which limits application in real-world scenarios as agents may need to be informed on the intentions of other agents in order to perform useful actions. For example, when autonomous vehicles negotiate a turn it is more useful for the vehicles to receive the action intention and state information of neighboring vehicles at the current time step, instead of aggregated past histories. The second shortcoming is that the encoding/decoding of communicated messages is dependent on the observation encoders of each agent, which eliminates the potential for explicit reasoning through a separate communication framework to lead to richer sets of behaviors and policies (as we show in this work).

Since CommNet, significant progress has been made in learning effective multi-agent communication (protocols) through the following methods: (i) broadcasting a vector representation of each agent's private observations to all agents [14, 17]; (ii) selective and targeted communication through the use of soft-attention networks [9] that compute the importance of each agent and its information [18, 16]; and (iii) communication through a shared memory channel [19, 20], which allows agents to collectively learn and contribute information at every time step. The architecture of [18] implements communication by enabling agents to communicate intention as a learned representation of private

observations, which are then integrated into the hidden state of a recurrent neural network as a form of agent memory. In contrast, Memory Driven Multi-Agent Deep Deterministic Policy Gradient (MD-MADDPG) [19] implements a shared memory state between all agents that is updated sequentially after each agent selects an action. However, the importance of each agent's update to the memory in MD-MADDPG is solely decided by its interactions with the memory channel.

Our work can be considered as an extension of MD-MADDPG [19], where SARNet instead works with independent memory states, and TarMAC [16], by computing the relevance of communicated information through query-key pairs of soft-attention [9]. However, our model extends these works by introducing the concept of relational learning through a two-step framework: generate relevance of the communicating agents through a modified query-key pair, and then specifically reason and attend over the *communicated information* and *past memories*.

The relational communication of our framework is based on the paradigm of relations in agent-based reinforcement learning, which was proposed by [11] through multi-headed dot-product attention (MHDPA) [9]. The core idea of relational reinforcement learning (RRL) combines inductive logic programming [21, 22] and reinforcement learning to perform reasoning steps iterated over entities in the environment. Attention is a widely adopted framework in Natural Language Processing (NLP) and Visual Question Answering (VQA) tasks [23, 24, 10] for computing such relations and interactions between entities. The mechanism [9] generates an attention distribution over entities, or more simply a weighted value vector based on importance for the task at hand. This method has been adopted successfully in state-of-the-art results for Visual Question Answering (VQA) tasks [23], [24], and more recently [10], demonstrating the robustness and generalization capacity of reasoning methods in neural networks.

## 3 Structured Attentive Reasoning Network

We introduce a communication architecture that is an adaptation of the attention mechanism of the Transformer network [9], and the structured reasoning process used in the MAC Cell [10]. The framework holds memories from previous time steps separately for each agent, which are then used for reasoning over new information received by communicating teammates. Agents learn relations between other entities or agents through a weighting mechanism, and consequently perform a reasoning operation over their memories and newly communicated information. This interaction between the structures of past memories and new information is used to produce a new memory for the agent that contains the most valuable information for the task at hand. The agent's new memory is then used to augment the encoded representation of the local observation $\mathbf{o}_i^t$ that is then used to predict the action of the agent.

To summarize, before any agent takes an action, the agent performs four operations via the following architectural features: (1) Thought Unit, where each agent encodes its local observations into appropriate representations for communication and action selection; (2) Question Unit, which is used to generate the importance of all information communicated to the agent; (3) Memory Unit, which controls the final message to be used for predicting actions by combining new information from other agents with an agent's own memory through the attention vector generated in the Question unit; and (4) Action Unit, that predicts the action. In Figure 1 we illustrate our proposed Structured Attentive Reasoning Network (SARNet).

### 3.1 The Thought Unit

The Thought unit at each time-step $t$ transforms an agent $i$'s private observations into three separate vector representations: query $\mathbf{q}_i^t$, key $\mathbf{k}_i^t$, and values $\mathbf{v}_i^t$, which are used for the Question and Memory units. The query $\mathbf{q}_i^t$ and key $\mathbf{k}_i^t$ are used to compute the relevance of communicated information in the Question unit, while the values $\mathbf{v}_i^t$ are used to integrate information into memory (Memory unit). Additionally, an encoding of the local observation, $\mathbf{e}_i^t$, is generated for the Action unit to complement the information generated from the communication step. Specifically, we have

$$\mathbf{q}_i^t, \mathbf{k}_i^t, \mathbf{v}_i^t, \mathbf{e}_i^t = \varphi_{\theta_i^{th}}^{th}(\mathbf{o}_i^t), \qquad \mathbf{q}_i^t, \mathbf{k}_i^t \in \mathbb{R}^{d_q}, \quad \mathbf{v}_i^t \in \mathbb{R}^{d_v}, \quad \mathbf{e}_i^t \in \mathbb{R}^{d_e} \qquad (1)$$

where $\mathbf{o}_i^t$ is the local observation of agent $i$, $\varphi_{\theta_i^{th}}^{th}$ can be characterized by a multi-layer perceptron (MLP) or any recurrent neural network (RNN) parameterized by $\theta_i^{th}$.

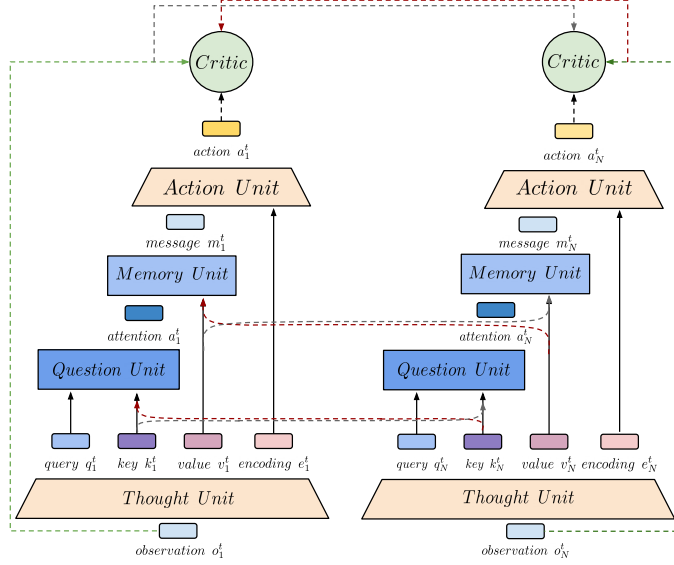

Figure 1: SARNet consists of a Thought unit, Question unit, Memory unit, and Action unit that process the agents observations and communications in distinct stages.

**The query** is used by each agent $i$ to inform the Question Unit of which aspects of communicated information are relevant to the current time step based on local observation.

**The key and value** are broadcast to all communicating agents. The key vector is used in the Question Unit to infer the relevance of the broadcasting agent to the current reasoning step, and the value vector is subsequently used to integrate the information into the memory of agent $i$.

The resulting information broadcasted by each agent $i$ to all the cooperating agents is then:

$$\mathbf{c}_i^t = [ \; \overbrace{\mathbf{k}_i^t}^{\text{key}} \quad \underbrace{\mathbf{v}_i^t}_{\text{value}} \; ]. \tag{2}$$

### 3.2 The Question Unit

The Question unit is designed to capture the importance of each agent in the environment, including the reasoning agent $i$, similar to the self-attention mechanism in [9]. In the mechanism used in [9], the attention computes a weight for each entity through the use of the dot-product computation and $softmax$. However, in contrast we use a linear projection instead of a dot-product to generate the attention mechanism over all individual representations in the vector for each entity, using Eq. 3. This allows the agent to compute the importance of each individual communicated information from other agents for a particular time step. More importantly, SARNet's use of a dedicated memory unit and the ability to simultaneously attend to both newly received information and past memories allows SARNet to have substantial performance gains over TarMAC, as TarMAC can only attend to new messages (values). This is performed through a soft attention-based weighted average using the query generated by agent $i$, and the set of keys, $\mathcal{K}$, that contains the keys $\{\mathbf{k}_1^t, \mathbf{k}_2^t, ..., \mathbf{k}_N^t\}$ from all agents.

The recipient agent, $i$, upon receiving the set of keys, $\mathcal{K}$, from all agents, including its own key, computes the interaction with every agent through a Hadamard product, $\odot$, of its own query vector, $\mathbf{q}_i^t$ and all the keys, $\mathbf{k}_j^t$, in the set $\mathcal{K}$. A linear transformation, $\mathbf{W}_{iq}^{[d_q \times d_1]}$, is then applied to every interaction, $\mathbf{qh}_{ij}^t$, that defines the query targeted for each communicating agent $j$, including self, to produce a scalar defining the weight of the particular agent:

$$\mathbf{qu}_{ij}^t = \mathbf{W}_{iq}^{[d_q \times d_1]}\mathbf{qh}_{ij}^t, \qquad \mathbf{qh}_{ij}^t = \mathbf{q}_i^t \odot \mathbf{k_j}^t, \qquad \mathbf{k}_j^t \in \mathbb{R}^{d_q}, \; \mathbf{qh}_{ij}^t \in \mathbb{R}^{d_q}, \; \mathbf{qu}_{ij}^t \in \mathbb{R}^1 \quad (3)$$

A $softmax$ operation is then used over the new scalars for each agent to generate the weights specific to each agent, i.e., an attention vector:

$$\mathbf{a}_i^t = \text{softmax} \left[ \frac{qu_{i1}^t}{\sqrt{d_q}} \dots \frac{qu_{ii}^t}{\sqrt{d_q}} \dots \frac{qu_{iN}^t}{\sqrt{d_q}} \right] \qquad \mathbf{a}_i^t \in \mathbb{R}^N \qquad (4)$$

The use of the linear transformation in Eq. 3 allows the model to specify an importance not only for each individual agent, as done in TarMAC [16], but further it learns to assign an importance to every element in the vector relative to each other in the information vector. Instead, TarMAC performs the equivalent step in Eq. 3 through a dot-product attention mechanism, which places equal importance to the elements in the query-key vectors to generate $\mathbf{qu}_{ij}^t$. We hypothesize that agents learn to attribute certain state information and intentions between elements of the query-key vectors. In lieu of the dot-product attention, we make use of a linear layer to compress the query-key Hadamard product to a weight, allowing more flexibility for the agent to learn varying representations of communication.

### 3.3 The Memory Unit

The Memory unit is responsible for decomposing the set of new values, $\mathcal{V}$, which contains $\{\mathbf{v}_1^t, \mathbf{v}_2^t, ..., \mathbf{v}_N^t\}$, into relevant information for the current time step. Specifically, it computes the interaction of newly communicated knowledge ($\mathcal{V}$) with the memory aggregated from the preceding time step. The newly retrieved information, from the memory and the values, is then measured in terms of relevance based on the importance of each agent generated in the Question unit.

As a first step, an agent computes a direct interaction between the new values from other agents, $\mathbf{v}_j^t \in \mathcal{V}$, and its current memory, $\mathbf{m}_i^{t-1}$. This step performs a relative reasoning between newly received information and the memory from the previous step. This element-wise multiplication allows the model to highlight relevant information from the prior memory, given the information from the new communications:

$$\mathbf{mi}_{ij}^t = \mathbf{m}_i^{t-1} \odot \mathbf{v}_j^t, \qquad \forall \, \mathbf{v}_j^t \in \mathcal{V}, \quad \mathbf{mi}_{ij}^t, \mathbf{m}_i^{t-1} \in \mathbb{R}^{d_v} \qquad (5)$$

The new interaction per agent $j$ evaluated relative to the memory, $\mathbf{mi}_{ij}^t$, and current knowledge, $\mathcal{V}$, is then used to compute a new representation for the final attention stage, through a feed-forward network, $\mathbf{W}_r^{[d_v \times d_v]}$. This enables the model to reason independently on the interaction between new information and previous memory, and new information alone:

$$\mathbf{mr}_{ij}^t = \mathbf{W}_r^{[d_{2v} \times d_v]} [\mathbf{mi}_{ij}^t + \mathbf{v}_j^t], \quad \mathbf{mr}_{ij}^t \in \mathbb{R}^{d_v} \qquad (6)$$

Finally, we aggregate the important information, $\mathbf{mr}_{ij}$, based on the weighting calculated in the Question unit, in (4). This step generates a weighted average of the new information, $\mathbf{mr}_{ij}$, gathered from the reasoning process, based on the attention values computed in (4). A linear transformation, $\mathbf{W}_m^{[d_v \times d_v]}$, is applied to the result of the reasoning operation to prepare the information for input to the Action unit:

$$\mathbf{mv}_i^t = \sum_{j=1}^N a_{ij}^t \mathbf{mr}_{ij}^t, \qquad \mathbf{m}_i^t = \mathbf{W}_m^{[d_v \times d_v]} \mathbf{mv}_i^t, \qquad \forall \, a_{ij}^t \in \mathbf{a}_i^t, \; \mathbf{m}_i^t \in \mathbb{R}^{d_v} \qquad (7)$$

### 3.4 The Action Unit

The Action unit, as the name implies, predicts the final action of the agent, $i$, based on the new memory, Eq. 7, computed from the Memory unit and an encoding, $\mathbf{e}_i^t$ (Eq. 1) of its local observation $\boldsymbol{o}_i$, from the Thought unit:

$$\mathbf{a}_i^t = \varphi_{\theta_i^a}^a(\mathbf{e}_i^t, \mathbf{m}_i^t), \qquad \mathbf{e}_i^t \in \mathbb{R}^{d_e}, \mathbf{m}_i^t \in \mathbb{R}^{d_v}, \mathbf{a}_i^t \in \mathbb{R}^{d_a} \qquad (8)$$

where $\varphi_{\theta_i^a}^a$ is a multi-layer perceptron (MLP) parameterised by $\theta_i^a$.

## 4 Experiments

We evaluate our communication architecture on Traffic Junction and OpenAI's multi-agent particle environment, [25], a two-dimensional stochastic environment consisting of agents and landmarks with

cooperative tasks. Each agent receives a private observation that includes only partial observations with limited or no vision of the environment depending on the task. We consider different experimental scenarios where a team of agents cooperate to complete tasks with static goals, or compete against other agents.

## 4.1 Baselines and Training

SARNet is compared against the communication architectures of TarMAC [16], IC3Net [15], Comm-Net [14] and a set of independent agents that do not communicate based on a recurrent-MADDPG (R-MADDPG) [25], [26]. In order to perform a fair and thorough analysis on the effectiveness of each communication architecture, *we adapt all of the baseline frameworks to the same recurrent model and training methodologies*. We use LSTM as observation encoders for the architectures. CommNet and TarMAC were originally proposed as models trained with global rewards, however, following the results of IC3Net on the strengths of training with individualized rewards, we adapt CommNet and TarMAC to be trained with independent rewards (see Appendix A.2 for details).

Additionally, we **propose a multi-agent recurrent extension of TD3** [12] with trajectory-based training and rollouts. The key contribution for training multi-agent systems: (1) Developing a training methodology with shared policy parameters and independent rewards. We use shared parameters for the actor, however, each agent maintains a distinct set of parameters for the critic to enable training with individualized rewards and centralized training with MADDPG [25]. (2) Extend R-MADDPG [26] from a single step transition update to a recurrent trajectory-based training [27] through TD3. For SARNet and all baselines, we use the above training methodology for multi-agent particle environments (MPE) [25], and REINFORCE [13] for traffic junction. Refer to Appendix A.1.3 for multi-agent trajectory-TD3. **Environment details** and additional experiments are described in Appendix B. Results for Physical Deception tasks are in Appendix B. **Hyperparameters** are noted in Appendix A.2.

## 4.2 Environments

Figure 2: Illustrations of the multi-agent particle environment in the experiments: Cooperative Navigation, Predator-Prey, Physical Deception, and Traffic Junction respectively.

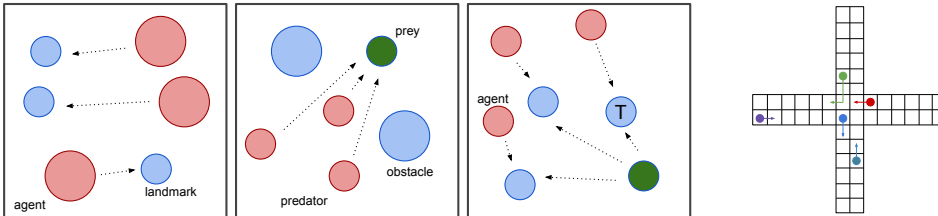

### 4.2.1 Cooperative Navigation

**Task**  In this environment, $N$ agents need to cooperate to reach $N$ landmarks. Each agent observes the relative positions of a fixed number of neighboring agents and landmarks. The agents are penalized if they collide with each other, and positively rewarded based on the proximity to the nearest landmark.

**Result Analysis**  In this task, SARNet agents consistently outperform the baselines, as shown in Table 1. In addition, SARNet maintains a low level of collisions with other agents. This behavior of SARNet agents is attributed to their ability to understand communicated information from other agents and effectively reason a course of action to stay away from a landmark that is heavily populated by other agents. We observe at the start of an episode, SARNet agents tend to gradually converge around their closest landmarks. However, if an agent observes that the landmark it is approaching is already occupied, SARNet agents do not force collisions with other agents. We also observe an interesting phenomenon of information overload in CommNet. As CommNet aggregates all messages irrespective of relevance, CommNet agents fail to identify the specific needs of communication and negotiation when maneuvering around highly occupied landmarks. IC3Net adopts a different

approach, where agents approach the nearest landmark and if there is a risk of collision with other agents, they tend to hover in place (which can be seen from high average distance to landmark values). TarMAC on the other hand is aggressive in its policy and tends to capture landmarks more consistently at the cost of collisions.

Table 1: Partially observable cooperative navigation. Number of collisions between agents, and average distance at the end of the episode are measured.

| Policy | $N = L = 6$ | | | $N = L = 10$ | | |
|---|---|---|---|---|---|---|
| | Reward | Coll. | Avg. dist. | Reward | Coll. | Avg. dist. |
| SARNet | **-12.39**$\pm$ **1.0** | 11.17$\pm$ 0.96 | 0.77$\pm$ 0.52 | **-14.73**$\pm$ **0.41** | 14.5$\pm$ 0.43 | 0.23$\pm$ 0.08 |
| TarMAC | -17.16$\pm$ 0.82 | 16.34$\pm$ 0.77 | 0.81$\pm$ 0.19 | -21.71$\pm$ 1.15 | 21.4$\pm$ 1.16 | 0.3$\pm$ 0.02 |
| CommNet | -23.34$\pm$ 5.25 | 22.51$\pm$ 5.13 | 0.81$\pm$ 0.13 | -21.31$\pm$ 2.47 | 21.03$\pm$ 2.46 | 0.28$\pm$ 0.03 |
| IC3Net | -16.97$\pm$ 1.17 | 13.62$\pm$ 0.20 | 3.35$\pm$ 0.98 | -27.67$\pm$ 0.55 | 27.35$\pm$ 0.58 | 0.28$\pm$ 0.01 |
| MADDPG | -13.12$\pm$ 0.80 | 12.11$\pm$ 0.75 | 1.01$\pm$ 0.19 | -20.97$\pm$ 0.15 | 20.82$\pm$ 0.11 | 0.14$\pm$ 0.02 |

**Communication Analysis**   We illustrate the communication patterns of SARNet and TarMAC in Figure 3, through their respective attention values. We observe that SARNet agents place importance on their own information the most, especially at the start of an episode where they need to make a decision on which landmark to approach. Attention to other agents is regulated mainly based on the proximity of other agents to the current target landmark. It is interesting to understand how differently SARNet and TarMAC utilize the attention mechanism. SARNet generally places a larger importance on agents that are gradually approaching, while TarMAC targets information from agents that are farther away. This key difference between the two communication architectures allows SARNet to outperform the baselines, by consistently maintaining a low collision rate. Additionally, as the agents stabilize to their target position, both SARNet and TarMAC attend equally to all agents, observing the intentions of other agents. We also investigate the specific impact of the memory unit in Appendix C.

### 4.2.2   Predator-Prey

**Task**   This task involves a slower moving team of $N$ communicating agents chasing $M$ faster moving agents in an environment with $L$ static landmarks. Each agent receives its own local observation, where it can observe the nearest prey, predator, and landmarks. We analyze the performance of predators trained with SARNet and the baselines while competing with preys trained with CommNet. The potential for communication and cooperation between the CommNet preys makes the task extremely hard for the predators.

**Results**   SARNet agents learn a sophisticated strategy of creating clusters and chasing the nearest prey, severely limiting the prey's potential for escape. Moreover, SARNet also often adopts the strategy to push a prey towards a landmark and surround it, forcing the prey into an adversarial state. SARNet successfully accomplishes a much better division of work as compared to TarMAC, in part due to actions being influenced by the communication at the same time step, allowing a better degree of approximation of other agents' policies and consequently their intentions. As for the non-communicating policy, MADDPG approaches the nearest visible prey in a naive manner, which is not always the optimal policy due to the nature of preys being faster than the predators. This results in MADDPG agents missing the target preys and needing to change course to re-target it. As the authors of IC3Net have noted, CommNet is equivalent to IC3Net, when trained with individualized rewards. In general, IC3Net requires a longer training time to enable the agents to learn to fully communicate in fully cooperative tasks, and that explains the general trend of IC3Net performing similarly to CommNet in a few tasks.

### 4.2.3   Traffic Junction

**Task**   Similar to [14], [15], [16] we perform an evaluation on Traffic Junction, with 6, 10 and 20 agents respectively. Agents enter the environment with a fixed probability $p_{arriv}$, with a preassigned route. The maximum number of agents in the environment are fixed, $N_{max}$. The agents can perform the following two actions: Gas or Brake. Once the agent completes the route it is sampled back into

Figure 3: **(Top)** Attention values generated by *Agent 1* in *red*, in cooperative navigation for $N = 6$ denoted by the shaded regions. Dashed lines indicate the normalized distance between *Agent 1* and other agents. **(Top-Left)** Attention metrics for SARNet. **(Top-Right)** Attention values generated by query-key pairs for TarMAC. **(Bottom-Left)** Learning curves for predator-prey environment with 6 predators trained with baselines, and 2 preys that learn with CommNet. Scores are measured by the cumulative rewards of predators and preys. SARNet consistently maintains a higher training score, measured as the difference in rewards between the predators and preys. **(Bottom-Right)** Mean success rates % across random seeds of the agents in the Traffic Junction task for 20 agents.

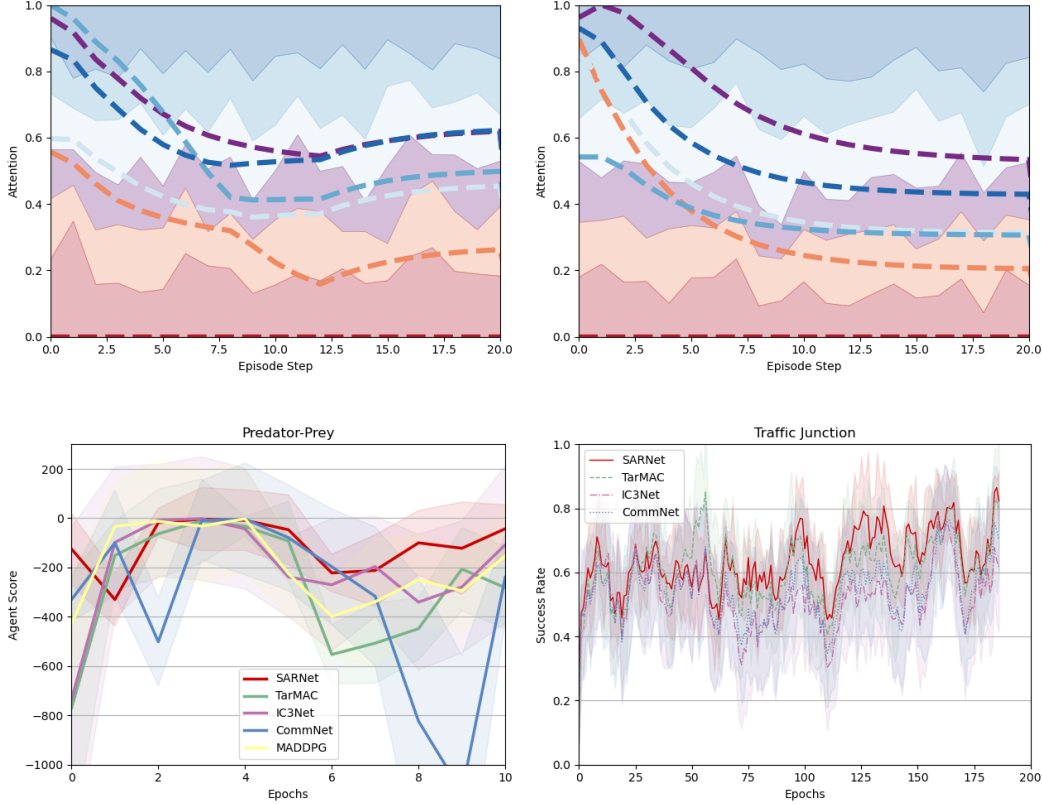

Table 2: **(Left)** Predator-Prey: Number of prey captures by the predators per episode is measured. SARNet performs better than all baselines. **(Right)** Success rate on traffic junction environments where agents have no vision of other agents.

| Predator vs Prey | 6 *vs* 2<br>**# captures** | 12 *vs* 4<br>**# captures** | Policy | $N = 6$ | $N = 10$ | $N = 20$ |
|---|---|---|---|---|---|---|
| SARNet vs. CommNet | **17.51**± **0.26** | **25.34**± **1.21** | SARNet | 74.27± 5.33 | **94.81**± **3.57** | **65.65**± **3.75** |
| TarMAC vs. CommNet | 16.18± 0.34 | 23.33± 3.38 | TarMAC | 76.35± 7.18 | 85.94± 5.10 | 59.36± 10.84 |
| CommNet vs. CommNet | 13.14± 0.24 | 18.83± 4.28 | CommNet | **77.08**± **5.02** | 83.58± 7.65 | 53.71± 5.57 |
| IC3Net vs. CommNet | 13.22± 0.28 | 17.16± 3.41 | IC3Net | 76.40± 4.64 | 89.82± 2.75 | 56.95± 9.30 |
| MADDPG vs. CommNet | 13.52± 0.36 | 15.33± 3.51 | | | | |

the environment. In order to make the task harder, the agents have zero visibility. This ensures that the task cannot be successfully solved without communication.

**Results** We train all the agents using REINFORCE [13] for the traffic-junction task, and report the results in Table 2. We observe that as the number of agents increase, SARNet has a significantly higher performance. Similar to TarMAC, SARNet learns to attend to other agents within its proximity, especially at the junctions. Since all of the architectures are trained using individualized rewards, the

architectures perform comparably for fewer agents. However, as the number of agents are increased, the task becomes significantly harder with high probabilities of collision and we clearly see the performance benefits of our architecture when communication is critical. SARNet inherently learns the action intentions of other agents during the current time step, instead of relying on the history of other agents, enabling it to coordinate much better than the baselines.

## 5 Conclusion

We have introduced a novel framework, SARNet, for communication in multi-agent deep RL which performs a structured attentive reasoning between agents to improve coordination skills. Through a decomposition of the representations of communication into reasoning steps, our agents outperform baseline methods in all tasks. Our experiments demonstrate key benefits of gathering insights from (1) an agent's own memories; and (2) the internal representations of the information available to an agent. The communication architecture is learned end-to-end, and is capable of computing task-relevant importance of each piece of communicated information from cooperating agents. While this multi-agent communication mechanism shows promising results, we believe that we can further adapt this method to scale to a larger number of agents, through the use of graph neural networks to initiate and maintain communication with indirectly connected agents, along with decentralized learning.

## Broader Impact

Multi-agent communication and cooperation is an active area of research that presents extreme challenges for agents to learn efficiently and in a scalable manner. Our work hopes to introduce a step towards learning sophisticated coordination strategies and behaviors in multi-agent learning and cooperation. Robotics has always been a natural application area for multi-agent learning. Communication in multi-agent research has varied applications, from the field of telecommunications, security and surveillance, to in fact bidding agents for Google Adwords [28]. Applications for deep reinforcement learning, especially in multi-robot systems, are still constrained due to the large sample complexity, and more importantly, coordination and cooperation are still left unsolved for larger tasks. Our work strives to achieve a robustness towards different learning mechanisms and tasks, while still learning to cooperate and communicate.

## Acknowledgements

This work was supported in part by NSF CNS-1830414 and NIFA 2018-67007-28380.

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
