[Supplementary Material]

# A Learning Multi-Agent Communication through Structured Attentive Reasoning: Appendix

## A.1 Policy Gradient Algorithms

Policy gradient (PG) methods are the popular choice for a variety of reinforcement learning (RL) tasks. In the PG framework, the parameters $\theta$ of the policy are directly adjusted to maximize the objective $J(\theta) = \mathbb{E}_{s \sim p^{\boldsymbol{\pi}}, a \sim \boldsymbol{\pi}_\theta}[R]$, by taking steps in the direction of $\nabla_\theta J(\theta)$, where $p^{\boldsymbol{\pi}}$, is the state distribution, $s$ is the sampled state and $a$ is the action sampled from the stochastic policy. Through learning a value function for the state-action pair, $Q^{\boldsymbol{\pi}}(s, a)$, which estimates how good an optimal action $a$ is for an agent in state $s$, the policy gradient is then written as, [13]:

$$\nabla_\theta J(\theta) = \mathbb{E}_{s \sim p^{\boldsymbol{\pi}}, a \sim \boldsymbol{\pi}_\theta}[\nabla_\theta \log \boldsymbol{\pi}_\theta(a|s) Q^{\boldsymbol{\pi}}(s, a)], \tag{9}$$

Several variations of PG have been developed, primarily focused on techniques for estimating $Q^{\boldsymbol{\pi}}$. For example, the REINFORCE algorithm [29] uses a rather simplistic method of sample return calculated as a cumulative expected reward for an episode with a discount factor $\gamma$, $R^t = \sum_{i=t}^{T} \gamma^{i-t} r_i$. When temporal-difference learning [30] is used, the learned function $Q^{\boldsymbol{\pi}}(s, a)$ is described as the *critic*, which leads to several different *actor-critic* algorithms [30], [31], where the actor could be a stochastic $\boldsymbol{\pi}_\theta$ or deterministic policy $\boldsymbol{\mu}_\theta$ for predicting actions.

### A.1.1 Partially Observable Markov Decision Processes

We consider a team of $N$ agents and model it as a cooperative multi-agent extension of a partially observable Markov decision process (POMDP) [32]. We characterize this POMDP by the set of state values, $\mathcal{S}$, describing all the possible configurations of the agents in the environment, control actions $\{\mathcal{A}_1, \mathcal{A}_2, ..., \mathcal{A}_N\}$, where each agent $i$ performs an action $\mathcal{A}_i$, and set of observations $\{\mathcal{O}_1, \mathcal{O}_2, ..., \mathcal{O}_N\}$, where each agent $i$'s local observation, $\mathcal{O}_i$ is not shared globally. Actions are selected through a stochastic policy $\boldsymbol{\pi}_{\theta_i} : \mathcal{O}_i \times \mathcal{A}_i \mapsto [0, 1]$ or through a deterministic policy $\boldsymbol{\mu}_{\theta_i} : \mathcal{O}_i \mapsto \mathcal{A}_i$ [33] with policy parameters $\theta_i$, and a new state is generated by the environment according to the transition function $\mathcal{T} : \mathcal{S} \times \mathcal{A}_1 \times ... \times \mathcal{A}_N \mapsto \mathcal{S}'$. At every step, the environment generates a reward, $r_i : \mathcal{S}' \times \mathcal{A}_i \mapsto \mathbb{R}$, for each agent $i$ and a new local observation $\mathbf{o}_i : \mathcal{S}' \mapsto \mathcal{O}_i$. The goal is to learn a policy such that each agent maximizes the total expected return $R_i = \sum_{t=0}^{T} \gamma^t r_i^t$ where $T$ is the time horizon and $\gamma$ is the discount factor.

### A.1.2 Deterministic Policy Gradient Algorithms

In the framework of deterministic policy gradient (DPG), the parameters $\theta$ of the policy, $\boldsymbol{\mu}_\theta$, are updated such that the objective $J(\theta) = \mathbb{E}_{s \sim p^{\boldsymbol{\pi}}, a \sim \boldsymbol{\mu}_\theta}[R(s, a)]$ and the policy gradient, (see section A.1), is given by:

$$\nabla_\theta J(\theta) = \mathbb{E}_{s \sim \mathcal{D}}[\nabla_\theta \boldsymbol{\mu}_\theta(a|s) \nabla_a Q^{\boldsymbol{\mu}}(s, a)|_{a = \boldsymbol{\mu}_\theta(s)}] \tag{10}$$

DDPG is an adaptation of DPG where the policy $\boldsymbol{\mu}$ and critic $Q^{\boldsymbol{\mu}}$ are approximated as neural networks. DDPG is an off-policy method, where experience replay buffers, $\mathcal{D}$, are used to sample system trajectories which are collected throughout the training process. These samples are then used to calculate gradients for the policy and critic networks to stabilize training. In addition, DDPG makes use of a target network, similar to Deep Q-Networks (DQN) [34], such that the parameters of the primary network are updated every few steps, reducing the variance in learning. Recent work [25] proposes a multi-agent extension to the DDPG algorithm, so-called MADDPG, adapted through the use of centralized learning and decentralized execution. Each agent's policy is instantiated similar to DDPG, as $\boldsymbol{\mu}_{\theta_i}(a_i|\mathbf{o}_i)$ conditioned on its local observation $\mathbf{o}_i$. The major underlying difference is that the critic is centralized such that it estimates the joint action-value $\hat{Q}(\boldsymbol{x}, a_1, ..., a_N)$, where $\mathbf{x} = (\boldsymbol{o}_1, \boldsymbol{o}_2, ..., \boldsymbol{o}_N)$. We operate under this scheme of centralized learning and decentralized execution of MADDPG, [25], as the critics are not needed during the execution phase.

### A.1.3 Trajectory-TD3 Updates

We extend the Twin Delayed Deep Deterministic Policy Gradient Algorithm for Markov Decision Processes, [12], to multi-agent systems and recurrent networks in partially observable domains (section A.1.1). We incorporate the centralized learning-decentralized execution framework from

[25], and the recurrent actor-critic from R-MADDPG [26], to learn the policy parameters for each agent. The R-MADDPG framework uses recurrent actor-critic models for training a multi-agent system, that can be seen as MADDPG with recurrent networks. However, the R-MADDPG framework fails to exploit the recurrent characteristics of their network during training time as they train using samples of a single time-step instead of employing a training mechanism through trajectories. The core idea of our proposed Multi-Agent Trajectory-TD3 (M-T2D3) updates are derived from the recurrent-DPG (RDPG) algorithm [27], which exploits trajectory based sampling and update rules for the actor and critic in a single agent case, and build upon it by incorporating the dual critic framework of TD3. We use the minimum target-Q values of both the critics across each time-step to compute the loss function for the critic $j$ updates for each agent $i$, given as:

$$\mathcal{L}(\psi_{ji}) = \frac{1}{NT} \sum_t \mathbb{E}_{\boldsymbol{x}, a_i, r, \boldsymbol{x}' \sim \mathcal{D}} \Big[ (Q_j^{\boldsymbol{\mu}_{\psi_i}}(c_t^{Qji}, \mathbf{x}_t, a_t^1, a_t^2, ..., a_t^N) - y_t^i)^2 \Big] \qquad (11)$$

$$y_t^i = r_t^i + \gamma \min_{j=1,2} Q_j^{\boldsymbol{\mu}'_{\psi_i}}(c_t^{Qji}, \mathbf{x}'_t, a_t'^1, a_t'^2, ..., a_t'^N)$$

where, $T$ is the length of trajectory sampled, $\mathbf{x'_t} = (\boldsymbol{o}_t'^1, \boldsymbol{o}_t'^2, \ldots, \boldsymbol{o}_t'^N)$ are the observations when actions $\{a_t^1, a_t^2, ..., a_t^N\}$ are performed, the experience replay buffer, $\mathcal{D}$, contains the tuples $(\mathbf{x_t}, \mathbf{x'_t}, \mathbf{m}_t, a_1, a_2, ...a_N, \mathbf{r}_t, \mathbf{c}_t^{Q1}, \mathbf{c}_t^{Q2}, \mathbf{c}_t)$, where $\mathbf{r}_t$ is $\{r_t^1, r_t^2, ..., r_t^N\}$ and, $\mathbf{m}_t, \mathbf{c}_t$ are the lists of recurrent hidden states of the communication memories and network. $\mathbf{c}_t^{Q1i}, \mathbf{c}_t^{Q2i}$ are the recurrent hidden states for the dual critics for each agent $i$. The target Q-value of agent $i$ is defined as $y_t^i$. We perform trajectory roll outs [35] for the recurrent hidden states, where only the first hidden state of the sampled trajectory, $i$, $c_t^{Qji}$ and $c_t^i$ and communication message $\mathbf{m}_t$,for each agent is used during both the actor and critic updates. The goal of the loss function, $\mathcal{L}(\psi_{ji})$ is to minimize the expectation of the difference between the current and the target action-state function. The gradient of the resulting policy, with communication, to maximize the expectation of the rewards, $J(\theta_i) = \mathbb{E}[R_i]$, can be written as:

$$\nabla_{\theta_i} J(\boldsymbol{\mu}_{\theta_i}) = \frac{1}{NT} \sum_t \mathbb{E}_{\mathbf{x}, a, \mathbf{m_i} \sim \mathcal{D}} \Big[ \nabla_{\theta_i} \boldsymbol{\mu}_{\theta_i}(\mathbf{c}_t, \boldsymbol{o}_t, \mathbf{m}_t) \times$$
$$\nabla_{a_i} Q_1^{\boldsymbol{\mu}_{\psi_i}}(c_t^{Q1i}, \mathbf{x}_t, a_t^1, \ldots, a_t^N)|_{a_t^i = \boldsymbol{\mu}_{\theta_i}(\mathbf{c}_t, \boldsymbol{o}_t, \mathbf{m}_t)} \Big] \qquad (12)$$

### A.1.4 Importance Sampling

We use a common replay buffer to store trajectories from all the parallel environments. However, each agent maintains a separate priority buffer to store and update its individual trajectory priority. We use a recurrent Prioritized Replay Buffer, that computes the importance of the stored trajectories as given in Recurrent Replay Distributed DQN (R2D2), [35]. Our replay prioritization is an extension of R2D2, where we use the minimum of temporal-difference (TD) errors $\delta_t^i$ for updating the prioritization values of each agent. The prioritization for each agent $i$ for each trajectory sampled is then given by

$$p^i = \eta \max_t \delta_i^i + (1 - \eta)\overline{\delta}_t^i$$
$$\delta_t^i = \min_{j=1,2} Q_j^{\boldsymbol{\mu}_{\psi_i}}(c_t^{Qji}, \mathbf{x}_t, a_t^1, a_t^2, ..., a_t^N) - y_t^i \qquad (13)$$
$$\overline{\delta}_t^i = \frac{1}{T} \sum_t \delta_t^i$$

### A.1.5 Training M-T2D3 for Individualized Rewards in Centralized Training

Due to the nature of centralized training, each agent is conditioned on the global state-action pair. For critics that share parameters between agents, each critic would receive a different reward for the same state-action pair, under which the agents fail to learn a policy. This necessitates the need to create distinct critics for each agent, that do not share all parameters. Although, in the interest of faster learning, it is possible to allow the initial encoders to share parameters, while leaving only the last layers as distinct parameters. This design enables the agents to be trained with different rewards for the same state-action sets. We maintain separate model instances for each agent's network, where the network only outputs the action of the individual agent. For the case of multi-agent

communication, all agents keep a copy of the other cooperating agents' encoders that are required for the communication process. Parameter sharing is used for each team of cooperating agents for actor network. Furthermore, as the agents maintain their own instances of networks, we can easily separate the information shared between competing agents, by not modeling the network of the competing agents within the agents graph. This is in contrast to the IC3Net framework of Individualized Reward Independent Controller (IRIC), where for centralized training, agents are required to maintain a copy of all cooperating and competing agents. The IC3Net framework essentially allows agents to *leak* information to other competing agents, and requires a *gating* function to control the *leak* of information. We adopt this framework for all mixed cooperative-competitive environments of Predator-Prey and Physical Deception for training SARNet and all the baseline architectures.

**Results**    We perform experiments on the cooperative navigation task with 6 SARNet agents using the following training methodologies,

1. R-MADDPG with single time-step transitions sampled from the replay buffer as described in [26].

2. TD3 with dual critics [12] extended to R-MADDPG with single time-step transitions.

3. R-MADDPG with RDPG [27] with updates made with trajectories of length 10 time-steps.

4. Multi-agent Trajectory-Twin Deep Deterministic Policy Gradient (M-T2D3), that is R-MADDPG with RDPG and dual critics of TD3, updated with sampled trajectories of length 10.

5. M-T2D3 with individualized importance sampling for each agent as described in Appendix A.1.3.

We observe a large performance improvement between R-MADDPG [12] and R-MADDPG with TD3. We note that the hidden states of the recurrent units that are sampled are from a trajectory generated by an *old* policy which introduces staleness in the updates. Since SARNet and other communication frameworks need to maintain a copy of each communicating agent's encoder, and consequently their hidden states, the issue of staleness during policy updates is further exacerbated as the network relies on all the hidden states of the communicating agent. Introducing a dual critic architecture with TD3 reduces the exploitation of the TD-errors and consequently a reduction in the variance of the updates. However, we still observe that the agents do not learn to exploit the dependencies between time-steps through the recurrent architecture, as both R-MADDPG and R-MADDPG with TD3 are trained with single time steps.

As we move the training to R-MADDPG with the trajectory based-RDPG [27], we see a drastic improvement in the performance of the network, as the recurrent network learns to utilize past information to significantly reduce the collisions using Backpropogation Through Time (BPTT). Extending the trajectory based updates to include TD3, we note an interesting phenomenon, where we observe that the training policy generates a conservative policy for the agents, by prioritizing the reduction in collisions at the expense of a larger average distance to the landmarks.

The use of Importance Sampling (IS), Appendix A.1.4, allows the policy to be updated through richer samples to learn effectively. We note a huge performance jump especially due to the fact that the environment is partially observable and the rewards for capturing the landmarks are sparse, due to additional constraints on avoiding collision between agents. Moreover as the number of training steps are limited due to time and hardware constraints, we observe that IS plays an important role in achieving a better policy when the training time is constrained.

## A.2    Training Details

**Individualized Rewards for Baselines**    Additionally, following the results of IC3Net [15], we adapt the framework of individualized rewards for training all the architectures. For the traffic junction task, since all agents cooperate with each other, we model the training graph with all the agents with shared parameters, and a single action and value head for each agent. Additionally, for mixed cooperative-competitive tasks such as Predatory-Prey and Physical Deception, we use the off-policy method of M-T2D3 for SARNet and the baselines, and model the agents according to our customised graph, refer Appendix A.1.5 to enable training through individualized rewards.

Table 3: Experimental results for partially observable cooperative navigation with 6 SARNet agents trained with different variations of DDPG.

| Policy | Reward | $N = L = 6$ Coll. | Avg. dist. |
|---|---|---|---|
| SARNET w/ R-MADDPG | -56.39$\pm$ 0.69 | 55.74$\pm$ 0.656 | 0.644$\pm$ 0.03 |
| SARNET w/ R-MADDPG + TD3 | -27.32$\pm$ 2.22 | 26.54$\pm$ 2.15 | 0.77$\pm$ 0.07 |
| SARNET w/ R-MADDPG + RDPG | -18.07$\pm$ 0.14 | 17.44$\pm$ 0.08 | 0.62$\pm$ 0.07 |
| SARNET w/ M-T2D3 | -17.52$\pm$ 0.85 | 16.65$\pm$ 0.31 | 0.85$\pm$ 0.09 |
| SARNET w/ M-T2D3 + IS | **-12.39**$\pm$ **1.0** | 11.17$\pm$ 0.96 | 0.77$\pm$ 0.52 |

**Training Policy**    We use batch synchronous method for off-policy gradient methods [36, 37] with M-T2D3 and individualized rewards for SARNet and all the baselines for the continuous action space task of Cooperative Navigation, Predator-Prey, and Physical Deception, Table 5. We train and benchmark all of our experiments for 3 random seeds. Benchmark results were reported after $80,000$ steps with 3 random seeds generated through the system clock. Our models were trained on an AMD Threadripper 3970x and Nvidia RTX 2080Ti, with Tensorflow 1.14. Training times for Traffic Junction were less than 12 hours, while for MPE environments the models were trained between 6 hours to 72 hours that scaled with the number of agents.

For the discrete action space task of Traffic-Junction we use REINFORCE to train for 3000 episodes, for SARNet and all baselines. Similar to the continuous action space tasks, all architectures shared similar network sizes, and hyperparameters described in Table 4, and A.2.

Table 4: Hyperparameters and Network Architecture

| Parameter | Details |
|---|---|
| Policy Encoder | LSTM-128 Units |
| Obs Pre Encoder | MLP-128 Units |
| Critic Encoder | GRU-256 Units |
| Action Encoder | MLP-128 Units |
| Key, Query Encoder | Linear-32 Units |
| Value Encoder | Linear-32 Units |
| Optimizer | ADAM |
| Actor Learning Rate | $10^{-3}$ |
| Critic Learning Rate | $10^{-3}$ |
| Actor Polyak ($\tau$, N agents) | $0.05/N$ |
| Critic Polyak ($\tau$) | 0.05 |
| Discount Factor ($\gamma$) | 0.96 |
| BPTT Length ($T$) | 10 Steps |
| Buffer Size | $10^5$ (10 steps each) |
| Critic Update Interval | 5 Steps |
| Actor Update Interval | 10 Steps |
| Parallel Environments | 200 (16 for REINFORCE) |
| $\alpha$ (priority) | 0.8 |
| $\beta$ (IS) | 0.6 |
| Total Training Steps (M-T2D3) | $5 \times 10^6$ |
| REINFORCE # Epoch's | 3000 |
| Memory Dropout | 0.85 |
| Read Dropout | 0.85 |
| Write Dropout | 1.0 |
| Output Dropout | 0.85 |
| Critic and Policy Regularization | $10^{-3}$ MSE |
| Exploration Noise | Gumbel-Softmax |

**Network Design for SARNet and Baselines**    We believe that architectures should be robust to different training methodologies, and hence we do not perform an extensive hyperparameter search for learning rate and other hyperparameters for SARNet or other baselines. Instead we use the most

Table 5: Comparison of various methods and their training methodology for continuous action space environments, not including Traffic Junction.

| Policy | Policy Update | Communication | Num. of Critics | Ind. Reward |
|--------|---------------|---------------|-----------------|-------------|
| SARNet | M-T2D3 | Scaled Attention | 2N | ✓ |
| TarMAC | M-T2D3 | Attention | 2N | ✓ |
| CommNet | M-T2D3 | Average | 2N | ✓ |
| IC3Net | M-T2D3 | Scaled Average | 2N | ✓ |
| MADDPG | M-T2D3 | None | 2N | ✓ |

common and widely used hyperparameters and model all the recurrent encoders as an LSTM unit. The authors of IC3Net propose their architecture with an LSTM, and additionally compare that to CommNet with an LSTM unit. TarMAC uses a Gated Recurrent Unit (GRU) in their original implementation and compare it to a CommNet modeled with a Vanilla RNN. Following IC3Net, we model all the communicating and non-communicating architectures with an LSTM unit, to reduce variance in results that might occur due to different choices of recurrent units.

## A.3 Structured Attentive Reasoning Network Algorithm

---

**Algorithm 1** Algorithm: SARNet with MT2D3

---

1: Initialize actors $(\boldsymbol{\mu}_{\theta_1}, \ldots, \boldsymbol{\mu}_{\theta_N})$ and critics networks $(Q_1^{\psi_1}, Q_2^{\psi_1}, \ldots, Q_1^{\psi_N}, Q_2^{\psi_N})$
2: Initialize actor target networks $(\boldsymbol{\mu}'_{\theta_1}, \ldots, \boldsymbol{\mu}'_{\theta_N})$ and critic target networks $({Q'}_1^{\psi_1}, {Q'}_2^{\psi_1}, \ldots, {Q'}_1^{\psi_N}, {Q'}_2^{\psi_N})$
3: Initialize common replay buffer $\mathcal{D}$ and separate importance buffer for each agent
4: **for** episode = 1 to E **do**
5:     Initialize a random process $\mathcal{N}$ for exploration
6:     Initialize the memory $\mathbf{m}_i$ for each agent
7:     **for** t = 1 to max episode length **do**
8:         **for** agent $i = 1$ to $N$ **do**
9:             Receive observation $\boldsymbol{o}_i$
10:            Generate query, key, value, encoding $\mathbf{q}_i, \mathbf{k}_i, \mathbf{v}_i, \mathbf{e}_i$, Eq. (1)
11:            Receive $K, V$ from all agents
12:            Compute the question vector $\mathbf{a}_i$, Eq. (4)
13:            Process the new information $\mathbf{m}_i$, Eq. (7)
14:            Store the new information in memory, $\mathbf{m}_i$
15:            Select action $\mathbf{a_i}$, Eq. (8)
16:        **end for**
17:        Set $\mathbf{x} = (\boldsymbol{o}_1, \ldots, \boldsymbol{o}_N)$ and $\boldsymbol{\Phi} = (\mathbf{m}_1, \ldots, \mathbf{m}_N)$
18:        Set $\boldsymbol{c} = (\boldsymbol{c}_1, \ldots, \boldsymbol{c}_N)$ where $\boldsymbol{c}$ is the hidden state of all recurrent encoders of agent $i$
19:        Execute actions $\mathbf{a} = (a_1, \ldots, a_N)$, observe rewards $r$ and next observations $\mathbf{x}'$
20:        Store $(\mathbf{x}, \mathbf{x}', \mathbf{a}, \boldsymbol{\Phi}, \mathbf{c}, r)$ in replay buffer $\mathcal{D}$
21:    **end for**
22:    **for** agent i = 1 to $N$ **do**
23:        Sample a random minibatch of $\Theta$ of $Z$ trajectories of length T $(\mathbf{x}, \mathbf{x}', \mathbf{a}, \boldsymbol{\Phi}, \mathbf{c}, r)$ from $\mathcal{D}$
24:        Set $y_t^i = r_t^i + \gamma \min_{j=1,2} Q_j^{\boldsymbol{\mu}'_{\theta_i}}(c_t^{Qji}, \mathbf{x}'_t, a_t'^1, a_t'^2, ..., a_t'^N)$
25:        Update critic by minimizing:
26:
$$\mathcal{L}(\psi_{ji}) = \frac{1}{ZT} \sum_t \mathbb{E}_{\boldsymbol{x},a_i,r,\boldsymbol{x}' \sim \Theta} \left[ (Q_j^{\boldsymbol{\mu}_{\psi_i}}(c_t^{Qji}, \mathbf{x}_t, a_t^1, a_t^2, ..., a_t^N) - y_t^i)^2 \right]$$

27:        Update Prioritized Replay Buffer, Eq. 13
28:        **if** episode mod 2 **then**
29:            Update actor with policy gradient, Eq. 12
30:        **end if**
31:    **end for**
32:    Update target networks:
$$\theta_i' = \tau \theta_i + (1 - \tau) \theta_i'$$
33: **end for**

---

# B Additional Environment Details

**Cooperative Navigation**

At each time-step, the agent receives an individualized reward of $-d$, where $d$ is the distance to the closest landmark, and penalized a reward of $-1$ for every collision that occurs with another agent. Moreover, the agents are also penalised $-0.01$ at every time-step if it hasn't captured a landmark. In this cooperative task, all agents strive to maximize their individual rewards. Performance is evaluated per episode by average reward, number of collisions, and average minimum distance to landmarks. Each agent's vision is limited to 4 landmarks and 3 other nearest agents. This environment is designed with individualised rewards.

**Predatory-Prey**

In this task, two teams of competing agents, predators and preys, work against each other. The task of the predator is maximize the number of touches on the prey within a time-step. The prey in turn needs to evade capture. Predators are rewarded by $+10$ every time they collide with a prey, and subsequently the prey is penalized $-10$. Since the environment is unbounded, the prey are also penalized for moving out of the environment. For the task, for $N = 6$ predators, and $M = 2$ preys, each agent can observe 1 predator, 1 prey, and 1 landmark. For the $N = 12$ and $M = 4$ task, the agents can observe 2 predators, prey and landmark each. Predators are rewarded $+10$ points for each capture of a prey, while the preys are rewarded $-10$ for every capture. Since, the environments are unbounded, we also penalize the preys for leaving the environment defined as, $r(x) = (x - 0.9) \times 10$ if $0.9 \leq x \leq 1$, $r(x) = 0$ if $x \leq 0.9$, else $r(x) = e^{2(x-1)}$. Both the predators and preys receive individualized rewards.

**Physical Deception**

**Task** A team of $M$ adversarial agents must close in on a target landmark, without directly observing it. The landmark must be inferred from the positions of $N$ communicating agents, whose task is to deceive the good agents. The adversarial agents must learn to spread out, such that the good agents fail to infer the target landmark. This task is highly complex and highlights the benefits of targeted attention, in SARNet and TarMAC compared to averaged-pooled attention of CommNet.

**Physical Deception** The communicating agents get positively rewarded by $d$ and negatively rewarded by $-d_a$, where $d$ and $d_a$ is the distance of the communicating and adversarial agent to the target landmark, respectively. This environment is treated as collaborative and the agents receive a shared global reward.

**Results** In this scenario, the adversary agents collaborate to seize all the landmarks, such that the good agents that are trained with MADDPG for $M = 1$ and CommNet for $M = 2$, cannot infer the target landmark. Due to complex collaboration required from the adversary agents, we observe that SARNet and TarMAC generates much more valuable information through its attention mechanism of query-key pairs, resulting in higher scores. In contrast, CommNet with its averaging technique performs poorly in $M = 1$ task as compared to MADDPG. We hypothesize that this might be due to unnecessary information that is being transmitted to the agent, which potentially acts as noise.

Table 6: For Physical Deception, we measure the avg. success rate across all of the communicating agents $N$, to reach the target landmark.

| Policy | $N = 4, M = 1$ Adv.success % | $N = 4, M = 2$ Adv.success % |
|--------|------------------------------|------------------------------|
| SARNet | $0.93 \pm 0.07$ | $0.81 \pm 0.02$ |
| TarMAC | $0.93 \pm 0.11$ | $0.79 \pm 0.01$ |
| CommNet | $0.75 \pm 0.01$ | $0.73 \pm 0.06$ |
| IC3Net | $0.82 \pm 0.11$ | $0.77 \pm 0.02$ |
| MADDPG | $0.88 \pm 0.03$ | $0.64 \pm 0.08$ |

**Traffic Junction** The agents are assigned individual rewards, shaped as a time penalty of $-0.01\tau_i$, where $\tau_i$ is the time taken for agent $'i'$ to complete it's route, and a collision penalty of $r_{coll} = -10$. We use the implementation of [15]. The tasks are modeled in an environment grid of $6 \times 6$, $14 \times 14$ and $18 \times 18$ grid for $6, 10, 20$ agents respectively. $p_{arrive}$ is set to 0.3, 0.02 and 0.02 respectively.

# C  Communication and Memory Analysis

## C.1  Analysis of Attention in Communication

We analyze the attention mechanism of computing weights through query-key pairs, first by the dot-product as described in Transformer [9] and used in TarMAC, and the linear projection based attention mechanism in Eq. 3. We analyze both the methods by applying them to the architecture of SARNet. We notice a general worsening of performance when the attention mechanisms are interchanged. This reinforces our initial hypothesis that SARNet benefits from a weighted reduction of the query-key pairs through a linear layer instead of a naive sum through a dot-product. We hypothesize that the linear projection allows the architecture to represent and condense the attention information in a more sophisticated manner to integrate the new information into memory.

Table 7: Experimental results for partially observable cooperative navigation with 6 SARNet agents with TarMAC's query-key attention mechanism.

| Policy | Reward | $N = L = 6$ Coll. | Avg. dist. |
|---|---|---|---|
| SARNet w/ TarMAC Attn | -32.56$\pm$ 2.42 | 30.77$\pm$ 2.28 | 1.77$\pm$ 0.25 |
| TarMAC | -17.16$\pm$ 0.82 | 16.34$\pm$ 0.77 | 0.81$\pm$ 0.19 |
| SARNet | **-12.39**$\pm$ **1.0** | 11.17$\pm$ 0.96 | 0.77$\pm$ 0.52 |

## C.2  Attention in Predator-Prey

Unlike the Cooperative Navigation task, where the task of the agents were to capture a static landmark, we analyze the attention values in a completely dynamic environment, where a team of 6 predators, chase 2 preys (CommNet agents). We analyze the attention of Agent 1 with respect to the proximity it has to each agent it attends to. As can be observed in Fig. 4, SARNet agents consistently group up while chasing a prey to increase the odds of capturing. On the other hand, TarMAC fails to learn this strategy, where the average distance of agent 1 to each predator is much higher throughout the episode. This strategy works in favor of SARNet as the preys are $50\%$ faster than the predators which significantly increases the difficulty of the task, which can be seen in Table 2. Moreover, the graphs also suggest that the distribution of attention over SARNet agents is more evenly spread out compared to that of TarMAC. As expected, in the scenario where an agent moves very far away (see agent in *peach* color at the 20*th* step), Agent 1 stops attending to it. This is due to the fact that Agent 1 and the two other agents in *green-yellow* and *teal* are already in pursuit to one of the preys, and the information of farther agents is not necessarily useful to Agent 1's immediate goals.

## C.3  Analysis of Memory in Communication

We study the performance of SARNet with varying memory size on the Cooperative Navigation task. Moreover, in order to truly understand the efficacy and impact of the reasoning based communication framework of SARNet, we also perform benchmarks where the agents can only use partial information from the memory, in Table 8. This is implemented during testing time, with a dropout layer with distinct *probabilities* of reading, writing and gathering information from the memory for action prediction. We observe a general worsening of the results as the dropout rate is increased, which is to be expected. More generally, as the dropout rate is increased, the agents' ability to read and write to the new memories decreases drastically, which significantly affects the ability of the agents to process information and complete the task. Specifically, we note that each agent's ability to read and write to the memory is crucial to the communication mechanism, and is an essential part of the agent's behavior in avoiding collisions.

Figure 4: Attention values generated by *Agent 1* in *red*, in predator-prey with 6 agents and 2 preys and landmarks each. Attention values are denoted by the shaded regions. Dashed lines indicate the distance between *Agent 1* and other agents. Attention values generated by query-key pairs **(Top)** for SARNet and **(Bottom)** for TarMAC.

Additionally, we showcase the dynamics of the memory channel in Fig. 5. We create a heatmap to analyze the information in the memory unit for different tasks where it visualizes the magnitude for each index in the memory unit. The communication patterns evolve as the dynamics of the agents change. We observe changes to the communication memory are more intense when coordination is imperative to the success of the task. This can be observed especially in the predator-prey task, where the memory unit is highly active due to the rapid changes in the coordinating agents' goals and consequently their behaviors. Alternatively, for the cooperative navigation task where all agents stabilize around landmarks within the first 15 to 20 time steps, it can be clearly seen in the heatmap of the memory unit that no further changes are written to the memory channel as the environment and agents become static. Through these observations, we believe that the introduction of a dedicated memory unit designed for communication is critical to each agent's decision making, and further substantiates our belief that it allows SARNet to perform better than other non-memory based architectures compared in this work.

Table 8: **(Left)** Metrics for the impact of memory size on SARNet performance for partially observable cooperative navigation for $N = L = 6$. **(Right)** Performance of SARNet when the architectures ability to integrate and read new information is reduced through the use of dropouts during testing time, where the percentage denotes the total ability to use the communication network.

| Policy | $N = L = 6$ Reward | Coll. | Avg. dist. | Policy | $N = L = 6$ Reward | Coll. | Avg. dist. |
|---|---|---|---|---|---|---|---|
| SARNet-8 | -27.08± 1.22 | 26.33± 0.65 | 1.04± 0.1 | SARNet-10% | -93.12± 0.54 | 79.09± 0.34 | 14.04± 0.88 |
| SARNet-16 | -46.13± 0.51 | 45.18± 0.49 | 0.95± 0.1 | SARNet-25% | -31.13± 1.26 | 30.06± 0.96 | 1.067± 0.3 |
| SARNet-32 | **-12.39**± **1.0** | 11.17± 0.96 | 0.77± 0.52 | SARNet-50% | -14.18± 0.19 | 13.34± 0.31 | 0.83± 0.11 |
| SARNet-64 | -19.43± 0.83 | 18.71± 0.76 | 0.73± 0.06 | SARNet-75% | -13.54± 0.23 | 12.61± 0.24 | 0.91± 0.14 |
| SARNet-128 | -24.27± 0.86 | 23.41± 0.25 | 0.86± 0.04 | SARNet-100% | **-12.39**± **1.0** | 11.17± 0.96 | 0.77± 0.52 |

Figure 5: We illustrate the *heatmaps* of SARNet's 32-bit memory unit of a single agent for the first 40 time-steps of an episode. **(Top)** for cooperative navigation with 6 agents, as the agents tend to stabilize around their final positions after 15 time-steps, we observe that no new information is being written to the memory. **(Bottom)** However, for predator-prey with 6 SARNet predators and 2 CommNet preys, the dynamic task requires information to be extracted and written to at every time step, as the state and actions of communicating predators, and competing preys change rapidly.