[Reviews · NeurIPS 2020]

Review 1

Summary and Contributions: The paper introduces a new multiagent framework which uses transformers and the MAC network to take a decision at each time step.The framework is divided into 4 stages where each stage is analogous to attention calculation in transformers. The framework also maintains a memory unit which allows the framework to combine the current information with information from the past. Results are shown on predator prey and traffic junction environments and SARNet is shown to perform better than the corresponding baselines.

Strengths: -The framework nicely decouples the action decision process into 4 steps which are sequential in nature. -SARNet consistently outperforms the rest of the baselines on multiple environments. -The supplementary has a great detail of analysis on what is important and what component affects the final performance by how much. -The paper also introduces an extension to the TD3 for trajectory based training in cooperative networks

Weaknesses: -The paper is overloaded with the division of framework into 4 units. The Thought unit probably can be combined with the Question unit for easier understanding as most of the paragraph only talks about how things will be used in the next units. -The experiments are not conducted on a larger scale for the Predator prey environment. -The framework is very similar to the TarMAC framework and I think this juxtaposition probably requires a paragraph of its own. To acknowledge the authors, they do compare with TarMAC dedicatedly at multiple levels but this digresses the comparison with other methods. I didn’t even see the discussion on the comparison with IC3Net. The paper seems to be very focused on distinguishing with TarMAC at some point but overall it doesn’t provide a clear juxtaposition. -Authors compare with IC3Net but they don’t mention if the framework can be extended to non-cooperative and competitive which is a speciality of IC3Net and a comparison is needed on that front. Experiments would be needed to verify the claim. -SARNet seems to work best on the medium scale of tasks. -Reasoning is talked about explicitly being present but there is no evidence of whether it actually happens quantitatively or qualitatively. I am not sure what authors mean by reasoning in the context of these environments. -Please see question to authors for rest of the comments.

Correctness: Yes, I didn't find any issues with that.

Clarity: The paper is well written, though some of the terms are overloaded and the division of the unit into 4 units can possibly be simplified.

Relation to Prior Work: The related work has been adequately covered but authors need to add more related work from recent conference publications.

Reproducibility: Yes

Additional Feedback: -Authors mention that all baselines have been adopted to use same methodology but it is not clear if the baselines also trained with extended TD3? -Physical deception task is talked about in experiments section but the actual results are present in supplementary. Please point to those or mention them as additional experiments in the main text Update: I thank authors for the rebuttal. I believe some of my concerns have been addressed but the core concerns still stands around similarity with TarMAC. Thus, I would like to keep my rating as it is and would like to see the paper get accepted given improvement in results. -Maybe I missed it, but in the introduction authors mention comparison against REINFORCE based baselines but I couldn’t find this comparison anywhere. -Line 150, individual has typo -The plots are not accessible and very hard to read. Please change the color scheme or stop over-laying similar colors on each other. - In intro, authors talk about using extended TD3 in competitive scenarios but no such experiment has been conducted.


Review 2

Summary and Contributions: The authors present a multi-agent communication strategy using a memory-based attention network: SARNet. The shared information by agents is processed by a Key-Query-Value attention mechanism similar to the Transformers [1]. Except that in SARNets, to compute the similarities between keys and queries a weighted dot-product (i.e, element-wise multiplication followed by a linear layer) is used instead of the standard one. Each agent has a decoupled memory unit, and the attention weights and values are used to aggregate this information into a single memory state which is used as an input to the policy at the current time-step. [1] Vaswani et al., Attention is all you need 

Strengths: The paper provides a good balance between novelty, sound experimental evaluation and reasonable improvements over existing competing methods. Although the novelty is mainly to make use of successful techniques (such as transformer like attention and use aggregated memory information from current time step to make decisions), I found the comparison to other baselines to be comprehensive; and the quantitative performance gains and behavioral differences on agents to be valuable to the NeurIPS community and worth of publication.

Weaknesses: I found some claims on the paper to be a bit misleading or at least lacking stronger evidence. (mainly the claims around reasoning, e.g l 225-228, since there was no analysis on what information is retained in memory). It's okay to include motivation for architectural choices, but one needs to provide empirical or theoretical evidence for stronger claims. Another weakness is the scalability of the method. The broadcasting of each agent information to all other agents will not be viable for a larger number of agents.

Correctness: Yes, I believe claims and method are correct.

Clarity: The paper is very well-written and flows nicely.

Relation to Prior Work: A few relevant missing citations: Jaderberg et al, 2019 Human level performance in first-person multiplayer games with population-based deep reinforcement learning. I am more familiar with the memory literature rather than multi-agent. In this work, authors emphasize that one of the differences from previous related work is to consider the aggregate memory information from the current step to make predictions rather than passing it to the next step. Note that this has been used on other memory focused models such as Graves et al., Hybrid computing using a neural network with dynamic external memory, 2016 Ke et al., Sparse Attentive Backtracking: Temporal Credit Assignment Through Reminding, 2018 Fortunato et al., Generalization of reinforcement learners with working and episodic memory, 2019 Ritter et al., Rapid Task-Solving in Novel Environments, 2020 Amongst others, so I suggest a bit of literature review on this area.

Reproducibility: Yes

Additional Feedback: **While the analysis on Appendix C on the role of the memory on the model provides some insights, perhaps reporting what happens if you remove attention all along would be more discerning. For example, the findings that if at test time you increase dropout rates on the memory component you have decay on performance are not surprising. This would probably happen if you do the same to other components of the architecture, since optimization by gradient descent will most likely not ignore them.   Finally, since there are claims about reasoning, exploring the effects of multi-step or multi-head attention would also be extremely valuable since most models focusing on memory/reasoning will use this. **Adding training curves with error estimates rather than only reporting final perfomance would strengthen the submission a lot since you can better compare baselines performances. ** An experiment on tasks with longer episodes would very valuable. From the plots, the longest episodes seem to have around 100 steps. Information could start degrading as episodes become longer, and perhaps more memory units per agent or some gating mechanism would be necessary. ** Is it possible to report estimates on running times for each task? Also, how many seeds did you use for your error estimates? I think it should be included in the manuscript.   ** I found the Bottom-Left plot on Fig 3 to be hard to read. Are the dashed lines the prey scores? Wouldn't plotting the difference in score be easier to read? Can you include error estimates on these plots? Bottom-right success rate plots look very similar for most baselines -- having error estimates would be of great help. ** In the introduction, using autonomous vehicles for motivating using current time step information for making decisions is tricky, since delays on sending and receiving these messages would impose a big challenge which this model doesn't address. I suggest finding a better example to motivate this approach. Few minor corrections: 1) Some of the citations are referred as arxiv submissions but are journal publications, so please update references (e.g [3], [10], [18], [24] -- which is also missing author information. 2) l. 245 two communication architecture --> architectures 3) any reason why you use two notations for keys/queries/value dimensions? Isn't Q=d_q, V=d_v? 4) Dimension of Wiq on eq 3 is incorrect. (as noted on appendix)   5) Algorithm 1 in A.3 line 23 (x, x', a, phi, r) --> (x, x', a, phi, c, r) (missing c) Updated review after authors response: I have read the rebuttal and I feel most of my concerns have been addressed. I am especially pleased to know an analysis for the memory unit will be included in the manuscript.


Review 3

Summary and Contributions: The paper proposes a communication architecture where agents (1) use an attention network to generate the relevance of the incoming messages and (2) reason over the received messages coupled with past memories. UPDATE: My main concern was this paper's similarity to TarMAC. However, I think this concern was adequately addressed in the author rebuttal. I am raising my score.

Strengths: + The paper presents detailed analyses that explain SARNet’s performance compared to baselines like TarMAC in the evaluation section. + The paper is clearly written and easy to follow.

Weaknesses: -SARNet seems very similar to TarMAC in terms of using attention. It would be helpful if the paper made the distinction between this paper and TarMAC clearer in Section 3 and Section 2. -Why does SARNet perform the worst in Traffic Junction with N=6 (when the task is easiest) and perform better with larger numbers of agents N=10 and N=20 (when the task becomes harder)? -Although SARNet clearly has the highest mean value in Table 2 (Right), the results do not seem very compelling, especially when looking at the Bottom-Right graph in Figure 3. Are there standard deviations that you could add to these bottom graphs?

Correctness: Yes

Clarity: Yes

Relation to Prior Work: For the most part. See my comments above wrt. TarMAC.

Reproducibility: Yes

Additional Feedback:


Review 4

Summary and Contributions: Paper presents a multi-agent reinforcement learning communication mechanism based on attention. The models allow agents to learn which information from observations of other agents that are relevant to communicate. The authors apply this model to a set of cooperative multi-agent environments. The model is compared with some recent multi-agent frameworks, more specifically TarMAC and CommNet. In the experiments the authors show that their presented method outperform previous work as well as simpler baselines.

Strengths: Paper presents convincing results that show that the presented method of perform recent multi-agent papers. Results are presented with standard deviations. The idea of selectively filtering what information from other agents, while not new, is an important area of research as larger multi-agent settings will require filtering to reduce computation and make the state space feasible for learning.

Weaknesses: The main weaknesses of this paper are: - Lack of comparison and reference to a very similar 2019 ICML paper "Actor-Attention-Critic for Multi-Agent Reinforcement Learning" by Iqbal et al. In that paper the authors also use attention to learn what observations are relevant to the agents. - Comparing RL algorithms can be difficult as RL is sensitive to hyperparameter choice and network design. As such using the same network and hyperparameters might disadvantage the baselines / compared method. Perhaps a better methodology would be to optimize baselines (in terms of network design, hyperparameters and training) independent given a rough budget of computation. - Figure 3 is very hard to decipher and gives little insight. Bottom right has very large overlaps between methods making it unclear how large the difference is. Bottom left has several series though it is unclear to me what the dashed lines represent. I'm not sure what the top row is supposed to present. Correlations between dashed lines and areas seems relatively low, maybe this plot should be represented some other way.

Correctness: Derivations and main claims are correct as far as I can tell, my only objection is the use of the same network and parameters for the baseline as mentioned in the weaknesses section.

Clarity: Paper is generally very well-written. Some sections and plots, most notably Figure 3 was difficult to read and decipher as noted above.

Relation to Prior Work: Generally well written prior works section. The big exception is the 2019 ICML paper "Actor-Attention-Critic for Multi-Agent Reinforcement Learning" that seems to be very similar and is not referenced or compared with.

Reproducibility: Yes

Additional Feedback:

[Author Response · NeurIPS 2020]

We would like to thank all of the reviewers for their time and thoughtful comments on our paper.

To begin, we concede that the attention mechanism in SARNet may appear similar to TarMAC (**Reviewers 1, 3**) and
related to Multi-Actor-Attention-Critic (MAAC) (**Reviewer 4**). However, there are important differences in SARNet
that we argue contribute to its significant gains in performance: (1) **how the attention mechanism is calculated**
**with respect to TarMAC**, and (2) **MAAC's use of critic-attention only to reduce state-space representation, not**
**for improving communicating policies**. TarMAC and MAAC both adopt dot-product attention that *equally* sums
across the query-key pairs [9]. In contrast, SARNet uses an attention scheme based on a Hadamard product followed
by a linear projection, which allows the network to generate richer and more effective communicating policies by
learning interactions *across* query-key pairs. To substantiate this claim, we performed an analysis of TarMAC's
dot-product attention applied to SARNet's memory in Appendix C.1, showing improvement in SARNet when moving
to a Hadamard/projection based attention. More importantly, SARNet's use of a dedicated memory unit and the ability
to *simultaneously attend* to both *newly received information and past memories* allows SARNet to have substantial
performance gains over TarMAC, as TarMAC can only attend to new messages (values). Regarding the **omission of**
**MAAC in our baselines**, our focus was on architectures that perform explicit communication during the execution
phase. **MAAC uses the attention mechanism for the centralized critic** during training and not in the action policy.
Based on **Reviewer 4**'s suggestion we will add results from **MAAC to complement MADDPG as a baseline without**
**communication**. Since receiving the reviews, we have performed initial evaluation with the following **results for**
**recurrent-MAAC with extended TD3 (MT2D3)**: (1) Cooperative Navigation ($N = L = 6$) resulted in an aggressive
policy with lower avg. distance to landmarks, but significantly higher collisions than SARNet, with rewards -22.02$_{\pm\ 0.87}$
*vs* SARNet's -12.39$_{\pm\ 1.0}$ , (2) Predator-Prey 6 *vs* 2 with a mean score of 14.49$_{\pm\ 0.46}$ *vs* SARNet's 17.51$_{\pm\ 0.26}$ .

With regards to our **training curves and attention metrics**, we agree with the reviewers and will improve the graphs
to make then more readable by **adding error bars in the training graphs** to better reflect training stability.

Our contribution of **MT2D3 has been applied to competitive scenarios in the paper, with Predator-Prey**, where
the agents compete with each other. We have described it in Appendix A.1.4 and we will add further details by including
figures on the design methodology. Agent training, **both for SARNet and all baselines**, was performed with **MT2D3**
**for the continuous action space environments**, and **REINFORCE for discrete tasks of Traffic Junction**.

**Reviewer 1**: We appreciate the feedback to make our paper more concise, and we **will combine the Thought and**
**Question Unit** in a single section. Choosing to have a **maximum of 20 agents for each environment** is attributed to
limits on computation and the fact that the baseline works in our paper have trained up to a maximum of 20 agents. For
Predator-Prey environments, we had a maximum of 12 *vs* 4 agents as training involves two different architectures with
different parameters, which heavily affects training time. We are actively working to address agent limits by introducing
a scalable multi-GPU multi-agent RL library to reduce training times, which will be released in the near future.

**Reviewer 2**: Your suggestion on including an analysis of the memory unit is very valuable. First, the term **reasoning** is
inspired from RRL [11] and NLP [24], where the authors term the interactions of query-key-value pairs as reasoning
between different entities. However, to clarify the reasoning that occurs in SARNet, we will add an **analysis of the**
**memory** through a Principal Component Analysis. Usage of **multi-step/multi-head attention** was explored, but it
required the memory unit's write method to use more computation time as it would require N-memory reads/writes.
SARNet can incorporate **forget/write gates for the memory unit for longer tasks** similar to that of an LSTM.
However, we did not see performance gains for the tasks in the paper. We will note results with *gates* in the revision. We
leave the **scalibility** of our approach for **larger tasks** for future work, through an extension with Graph Neural Networks.
**Estimates on running times** for tasks are reported in Appendix A.2, and will be noted in a dedicated table. We agree
with the reviewer, and will revise the manuscript to add a **descriptive analysis of IC3Net**. As the authors of IC3Net
have noted, IC3Net is CommNet with gates when trained with individualized rewards. The additional complexity in
training of the gating function in cooperative environments partially explains IC3Net's lower performance.

**Reviewer 3**: We have described **key differences between SARNet and TarMAC** in our response (lines 2-13). Ad-
ditionally, SARNet is equipped with a distinct memory unit that does not rely on an RNN encoder to aggregate
messages, and is thus *adaptable to non-recurrent observation encoders*. **Performance of SARNet in Traffic Junction**
**for 6 agents** is within the standard deviations of the baselines as communication is not critical for a few agents.
However, SARNet's performance is substantially better than baselines when the task becomes harder (more agents) and
communication is key, a trend that can be observed across all environments.

**Reviewer 4**: The suggestion to include MAAC as a baseline is highly appreciated, and we will include it as part of the
baselines, along with extending SARNet with MAAC. We address our original motivation for our baseline selection on
lines 13-20 in our response. **Hyperparameters** were carefully chosen over 10 test runs to accommodate near-optimal
learning, and originally proposed networks sizes for all architectures. Additionally, we agree with the suggestions to
**improve the figures**, which is addressed in lines 21-22 in our response.

[Meta-Review · NeurIPS 2020]

All 4 reviewers suggest that this paper is above the acceptance threshold. After reading the reviews and author response, I will recommend acceptance with a note (see bellow). Reviewers agree that the idea of filtering what information from other agents to consider within a multi-agent setup, while not new, is an important area of research, especially as the number of agents and the complexity of the environments grows. All reviewers agree that this is a sound submission, with good experimental results and comprehensive comparisons with previous work. Some concerns were raised, but the author response did a good job in addressing many of those. NOTE TO AUTHORS: Overall, all reviewers have raised very constructive and thoughtful comments, and I would like to see their points addressed in the final manuscript (basically incorporate elements of the author response in the final camera ready). In particular, the discussion about similarity to other methods, updating the result with training curves and attention metrics.